# MemOracle: Symbolic Reasoning with Retrieval for Harmful Content Detection

## Abstract

Memes, as a prevalent form of online communication, often combine text and imagery to convey complex and sometimes harmful messages. Detecting hateful content in memes poses significant challenges due to their multimodal nature and the requirement for contextual reasoning. We propose a novel framework built upon vision-language models, enhanced with multimodal retrieval and symbolic reasoning, to assess the harmfulness of memes. Specifically, our system first parses the input meme using a vision-language model to extract image-text elements and semantic descriptions. These are embedded into a joint representation and stored in a vector database. For any given query meme, similar examples are retrieved from the database. A large language model is then employed to reason over the query meme in light of the retrieved examples, guided by a predefined definition of hateful memes and a symbolic Chain-of-Thought prompt. The reasoning proceeds in three stages: translator, planner, and solver, producing both a decision and an explanatory rationale. Our approach enables a more transparent and context-aware assessment of online multimodal content. Comprehensive experiments on the public FHM, HarM, and MultiOff datasets demonstrate that MemOracle consistently surpasses state-of-the-art hateful meme detection models in terms of accuracy, balanced accuracy, and Matthews correlation coefficient, highlighting its effectiveness in interpreting and identifying harmful content. The code is available at `https://anonymous.4open.science/r/MemOracle-C66F/README.md`

## 1 Introduction

Memes have emerged as a powerful form of online expression, combining visual elements with text to convey complex, often culturally embedded messages. Beyond their humorous or entertaining purposes, memes can also serve as a vehicle for covertly spreading hateful ideologies, posing a serious threat to a healthy and inclusive online environment. Detecting hateful memes (Lippe et al., 2020; Kiela et al., 2020; Cao et al., 2020) is particularly challenging because harmfulness often lies not in explicit surface features, but in implicit semantic associations, cultural references, or sarcasm. Correctly identifying such content requires not only multimodal understanding, but also a reasoning process capable of inferring hostility from subtle, indirect cues.

Consequently, hateful meme detection involves an inference challenge beyond simple feature matching, requiring logical reasoning chains to uncover hidden intent embedded in visual–textual combinations. Chain-of-Thought (CoT) reasoning, which generates step-by-step natural language rationales before producing the final answer, has achieved notable success in various natural language processing tasks (Wei et al., 2022; Zhang et al., 2023; Fei et al., 2023; Inaba et al., 2023). Building on this, `Evolver` (Huang et al., 2025) attempts to enhance detection performance by incorporating Chain-of-Evolution prompts, while `MinD` (Liu et al., 2025) further proposes a multi-agent debate mechanism to ensure robust decision-making through reasoned arbitration. Despite these advances, existing approaches still face two key limitations: first, the absence of structured symbolic logic constrains interpretability; second, the single-instance inference paradigm is hindered by insufficient contextual information, resulting in limited performance in detection tasks.

To this end, we propose MEMORACLE, a framework specifically designed to address the challenge of hateful meme detection. The name MEMORACLE derives from three inspirations: meme, memory, and oracle. It highlights the module's role as a knowledge oracle that retrieves relevant historical meme instances from memory to guide the interpretation of new memes. This framework integrates a symbolic logic-driven reasoning layer to enable deeper semantic understanding and explainability. In addition, an associative retrieval module, to further enhance contextual comprehension. Unlike conventional neural CoT pipelines that lack structured logic, our symbolic reasoning layer supports internal logical chaining, promoting both transferability across tasks and interpretability. Associative retrieval equips the model with the ability to retrieve relevant historical meme instances from memory, helping the system learn meme construction patterns and align current content with prior hateful expressions. Hence, compared to conventional methods, MEMORACLE can more effectively uncover the implicit, context-sensitive hostility embedded in memes, particularly those relying on cultural knowledge, sarcasm, or intertextual references.

We conducted extensive experiments on four public hateful meme datasets, including FHM, Multi-Off, HarM-c, and HarM-p, demonstrating state-of-the-art performance across all benchmarks. Compared to vanilla multimodal models and existing detection frameworks, our method significantly improves detection accuracy (ACC), balanced accuracy (BACC), and Matthews Correlation Coefficient (MCC). Our main contributions are summarized as follows:

- We introduce structured symbolic reasoning in the context of hateful meme detection, effectively enhancing both the interpretability and performance of the detection.
- We propose MEMORACLE, a retrieval-augmented associative module that leverages historical meme knowledge to overcome the information bottleneck inherent in single-instance inference.
- Extensive experiments on multiple hateful meme datasets demonstrate that our method achieves state-of-the-art performance, outperforming previous frameworks MinD and Evolver by up to 9% in the detection task.

## 2 RELATED WORK

This section reviews existing harmful meme datasets and related detection methods, serving as a background and reference for the design of subsequent approaches.

### 2.1 HARMFUL MEME DATASETS

The underlying harmful content in memes poses a significant threat to the online ecosystem. In response, multiple datasets have been established. Kiela et al. (2020) introduced the Facebook Hateful Memes dataset (FHM) as a challenge set focusing on the detection of hateful speech in multimodal memes, designed to emphasize genuine image–text reasoning through benign confounders. Suryawanshi et al. (2020) developed a multimodal meme dataset (MultiOff) for offensive content, which focuses on memes associated with the 2016 US presidential election, and captures authentic political sarcasm. Pramanick et al. (2021a) presented a harmful memes dataset (HarM), which encompasses COVID-19-related memes, and subsequently extended the dataset to include U.S.-politics-related memes (Pramanick et al., 2021b). Fersini et al. (2022) proposed the Multimedia Automatic Misogyny Identification dataset (MAMI), which is designed for the detection of misogynistic memes. More recently, Lu et al. (2024b) constructed TOXICN-MM, a Chinese dataset for harmful meme analysis. In our study, we employ FHM, MultiOff, and HarM to construct a balanced benchmark that spans both challenge-style and real-life memes, enabling a robust evaluation of both accuracy and generalization.

### 2.2 HATEFUL MEME DETECTION

Earlier studies typically used two-stream models to integrate visual and textual information (Kiela et al., 2020; Suryawanshi et al., 2020; Lippe et al., 2020). These models relied on large-scale annotated data and were limited in providing logical explanations for multimodal inputs. The mainstream approach has been to employ multimodal fusion techniques and attention-based mechanisms to distinguish harmful memes. Kiela et al. (2020) cast hateful-meme detection as a binary classification

task based on each meme's image plus OCR text, comparing unimodal baselines to early-fusion multimodal models that require genuine image–text reasoning. Pramanick et al. (2021b) classified harmful memes with a multimodal network, attributing cues through intra-modal and cross-modal attention, while predicting harmfulness through multitask heads with focal loss. Another method leverages pretrained vision-language models (VLMs) fine-tuned for meme classification. Lippe et al. (2020) fine-tuned early-fusion transformers and improved performance with confounder up-sampling, loss reweighting, and cross-validated ensembling to classify hateful memes. Hee et al. (2022) fine-tuned VLMs for meme classification and subsequently examined the classifiers with gradient-based attribution and attention-grounding analyzes. More recent work explored the broader use of large multimodal models (LMMs), highlighting their ability to operate in a zero-shot setting without task-specific supervision. Huang et al. (2025) prompted an LMM in a zero-shot manner by retrieving similar memes, summarizing shared harmful cues, giving a brief task definition, and then classifying through in-context prompting. Liu et al. (2025) constructed a zero-shot multi-agent pipeline that retrieves similar memes, derives bidirectional interpretations, and then runs a debate-and-judge step to reach a robust decision. However, existing VLM-based methods lack logical reasoning in the processing of multimodal content, which limits the interpretability and effectiveness of the results. Our proposed MEMORACLE framework integrates symbolic reasoning based on first-order logic (FOL) to strengthen the inference process, together with associative retrieval that leverages meme analyzes for semantic-level recall. This combination improves the robustness, contextual awareness, and interpretability of multimodal hateful meme detection.

# 3 METHOD

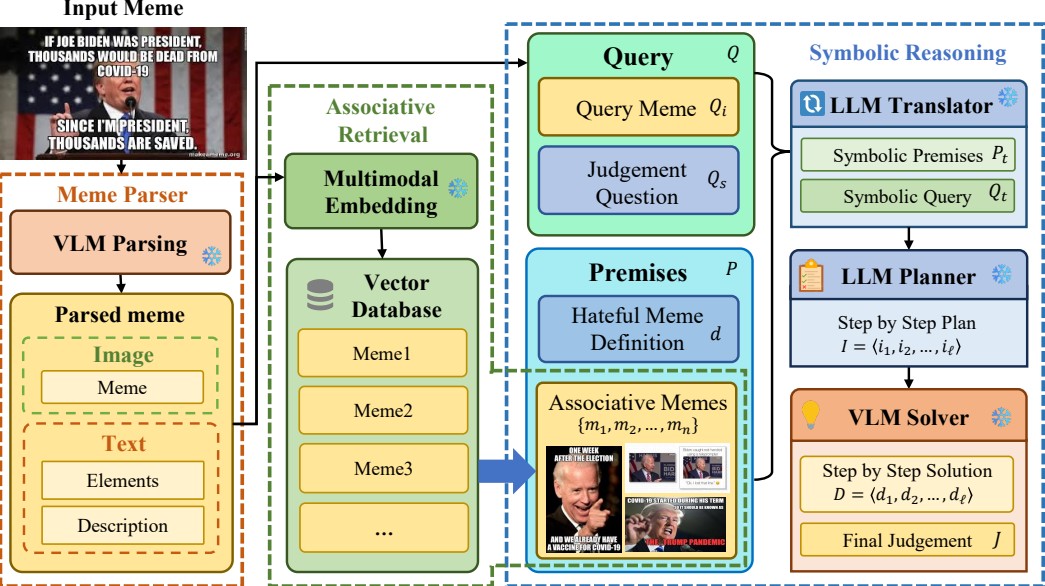

Figure 1: Framework of MEMORACLE

## 3.1 MEMORACLE FRAMEWORK

Our proposed MEMORACLE framework detects potentially harmful memes by integrating multimodal understanding, associative retrieval, and symbolic reasoning. As shown in Figure 1, the framework consists of three main stages:

**(1) Meme Content Parser:** The system starts with a multimodal parsing module based on a VLM, which jointly interprets visual content and overlaid text. This stage extracts the key semantic elements of the meme and produces a descriptive summary, serving as the initial representation for subsequent processing.

**(2) Associative Retrieval.** The parsed representation is embedded in a shared multimodal space derived from CLIP and compared against a stored vector database of meme embeddings. Semantically related memes are retrieved to construct a contextual set of reference cases. Together with the dataset definitions of hateful content, these retrieved memes are organized as *Premises*, while the target meme serves as *Query* for downstream reasoning.

**(3) Symbolic Reasoning.** Given the Query and the Premises, a structured reasoning pipeline is applied, consisting of three stages— *Translator*, *Planner*, and *Solver*. The Translator converts multimodal input into symbolic form; the Planner assembles candidate reasoning chains; and the Solver executes the chain to produce a final decision and an explanatory rationale on whether the meme is harmful or not. The subsequent subsections provide details of each stage.

## 3.2 Vision-Language Model Meme Parsing

As shown in Figure 2, the first stage of MemOracle processes each meme image using a pre-trained vision-language model (VLM), which jointly parses its visual and textual elements. Specifically, the model detects salient visual entities (e.g. "a crying human face", "cartoon characters") and text overlays (e.g. meme captions, speech bubbles) and then generates a description that captures the semantic content of the scene. The description typically identifies the entities involved, their actions, and relevant emotional or contextual cues inferred from both image and text. The output serves as the initial semantic representation of the meme. This semantic representation not only mitigates the noise from low-level appearance variations but also provides a more robust input foundation for subsequent structured reasoning.

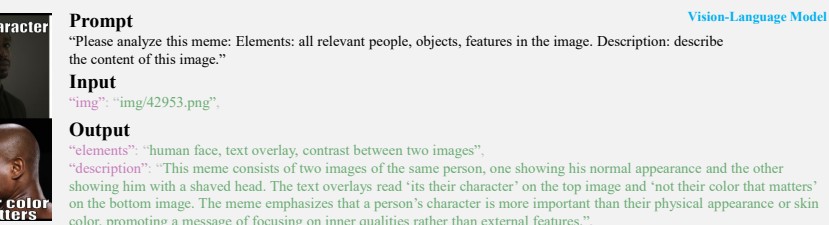

Figure 2: Visualization of the Parser step

## 3.3 Multimodal Embedding and Associative Retrieval

To enable semantics-oriented retrieval, each parsed meme is projected into a shared multimodal embedding space. We adopt CLIP to encode three complementary sources of information: the raw image, the extracted elements, and the generated description. Their embeddings, denoted as $\mathbf{v}_{\text{img}}$, $\mathbf{v}_{\text{elem}}$, and $\mathbf{v}_{\text{desc}}$, are combined with weighted importance:

$$\mathbf{v}_{\text{final}} = \lambda_0 \cdot \mathbf{v}_{\text{img}} + \lambda_1 \cdot \mathbf{v}_{\text{elem}} + \lambda_2 \cdot \mathbf{v}_{\text{desc}} \tag{1}$$

where $\lambda_0$, $\lambda_1$, and $\lambda_2$ are tunable hyperparameters controlling the relative importance of the image, elements, and generated description embeddings, with $\lambda_0 + \lambda_1 + \lambda_2 = 1$, ensuring a convex combination of the three components. The resulting vectors are stored in a Milvus vector database, forming a unified semantic representation of the meme corpus. The database is continually updated as new memes are encountered, supporting scalability and adaptability.

Given a new query meme, we apply the above parsing and embedding procedure to obtain its embedding vector $\mathbf{r}_q$. Let $\{\mathbf{r}_1, \ldots, \mathbf{r}_N\}$ denote the embeddings of all memes in the database, corresponding to memes $\{m_1, \ldots, m_N\}$. The similarity between the query and each database embedding is measured using cosine similarity:

$$s_i = \cos(\mathbf{r}_q, \mathbf{r}_i), \quad i = 1, \ldots, N \tag{2}$$

The top-$K$ most similar memes are then retrieved as:

$$\mathcal{R} = \{m_i \mid s_i \text{ is among the top-}K \text{ values of } \{s_1, \ldots, s_N\}\} \tag{3}$$

This semantic retrieval allows the system to identify memes that are conceptually related, even when their visual or textual surface forms differ significantly. The retrieved set $\mathcal{R}$ then serves as contextual premises for subsequent symbolic reasoning.

## 3.4 SYMBOLIC REASONING

The goal of this stage is to integrate the information from the previous two stages and perform structured symbolic reasoning to yield an interpretable judgment on whether the query meme $Q = (Q_i, Q_s)$ is harmful where $Q_i$ denotes the parsed meme and $Q_s$ denotes the judgment query (e.g. "Is this image harmful?"). Formally, given a set of premises $P = \{d, m_1, m_2, \ldots, m_n\}$ where $d$ denotes the dataset-provided definitions of hateful content and each $m_i$ is a retrieved meme.

The reasoning pipeline unfolds in three stages: **(1) Translator:** converts multimodal meme representations into structured first-order logic(FOL) symbolic forms that explicitly capture entities, relations, and contextual cues; **(2) Planner:** constructs a logical reasoning chain that connects the premises $P$ with the query $Q$, ensuring step-by-step interpretability; **(3) Solver:** in the final step, the original visual modality $Q_i$ is incorporated, and a VLM serves as the inference engine. This enables the system to preserve visual evidence during reasoning, thereby mitigating the information loss incurred when visual signals are abstracted into textual symbols.

By coupling symbolic reasoning with multimodal cues, the framework produces judgments that are both logically consistent and faithful to the original multimodal content.

### 3.4.1 TRANSLATOR

**Prompt** Large-Language Model
"Please translate the following information into **First-Order Logic (FOL)** expressions, in a structured and concise format suitable for automated reasoning."

**Input**
"Premise":{ "dataset_content": "A direct or indirect attack on people based on characteristics...",
"associative_img1": "The meme contrasts two men, one in a suit and the other bald, suggesting that their character, rather than their physical appearance, is what truly matters...",
"associative _img2": "The meme humorously comments on the unexpected twist in a porn video plot, where a character initially identified as a transgender person is revealed to be a midget...",
...}
"Query":{ "Query Image": "The meme contrasts two men, one in a suit and the other bald, suggesting that...",
"question": "Does this image contain hate speech?"}

**Output**
"Symbolic Definition": ["Attack(x): x is an attack based on characteristics", "Violent(x): x is violent speech or actions", "Dehumanizing(x): x compares people to non-human things"]
"Translated Premise": { "dataset_content": [ "∀x (Attack(x) ↔ (Violent(x) ∨ Dehumanizing(x) ∨ Inferiority(x) ∨ Exclusion(x) ∨ Segregation(x) ∨ MockingHateCrime(x)))..."],
"related_img1": ["∀x (TwoPanelLayout(x) ∧ Man(x) ∧ WearsSuit(x) ∧ Bald(x) ∧ TextTop('its their character', x)..."],
"related_img2": ["∀x (SinglePanelLayout(x) ∧ Man(x) ∧ WearsSuit(x) ∧ HoldsMicrophone(x) ∧ Smiling(x)..."]}
"Translated Statement": {"query_meme": {"Man(x) ∧ WearsSuit(x) ∧ Man(y) ∧ Bald(y)..."},
"question": "ContainsHateSpeech(x)"}

Figure 3: Illustration of the Translator process. Visualizations of the other reasoning steps are provided in the Appendix.

As shown in Figure 3, the Translator converts the premises $P$ and the query $Q$ from natural language into a symbolic format suitable for logical reasoning. This ensures that the meme content, which encompasses visual cues, textual statements, and contextual references, is uniformly represented for the downstream modules.

Formally, the premises are mapped into first-order logic (FOL) as:

$$P_t = \{d_t, m_{1t}, m_{2t}, \ldots, m_{nt}\}, \tag{4}$$

where the subscript $t$ denotes the translated symbolic form. Similarly, the query meme is translated into its symbolic representation: $Q_t = \{Q_{it}, Q_{st}\}$, which serves as input to the Planner.

### 3.4.2 PLANNER

The Planner generates an interpretable, step-by-step reasoning plan that connects the premises to the query, producing explicit instructions for downstream execution. To mitigate information loss

| Method | FHM | | | Multioff | | |
|---|---|---|---|---|---|---|
| | ACC | BACC | MCC | ACC | BACC | MCC |
| **Open-source LMM (Zero-shot)** | | | | | | |
| LLaVA-1.5-13B | 55.01 | 55.13 | 10.65 | 48.15 | 56.23 | 14.41 |
| Qwen2.5VL-7B | 63.20 | 63.01 | 26.53 | 64.43 | 59.94 | 21.70 |
| LLaVA-v1.6-Vicuna-13B | 56.90 | 56.66 | 13.72 | 61.07 | **65.94** | 33.46 |
| LLaVA-OneVision-Qwen2-7B | 56.60 | 56.25 | 13.37 | 59.73 | 60.16 | 19.81 |
| InternVL3.5-8B | 60.40 | 60.36 | 20.81 | 62.42 | 62.04 | 23.57 |
| DeepSeek-VL-7B | 54.35 | 54.31 | 8.63 | 61.49 | 61.80 | 22.97 |
| **Reasoning Based Methods** | | | | | | |
| MinD | 63.80 | 64.21 | 31.13 | 61.74 | 64.30 | 28.30 |
| Evolver | 59.60 | 59.08 | 21.32 | 65.10 | 58.92 | 21.70 |
| MemOracle | **66.10** | **65.84** | **32.83** | **69.80** | 65.58 | **34.03** |

Table 1: Results on the binary classification datasets FHM and MultiOff.

from pure symbolic abstraction, the Planner operates on a merged representation that combines both natural language and symbolic views. Let $\oplus$ denote a merge operator (e.g., concatenation or structured fusion); we define

$$P_c = P \oplus P_t, \qquad Q_c = Q \oplus Q_t. \tag{5}$$

Conditioned on $(P_c, Q_c)$, the Planner produces a sequence of intermediate reasoning steps.

$$I = \langle i_1, i_2, \ldots, i_\ell \rangle \tag{6}$$

where each $i_j$ corresponds to a concise and human-readable instruction that applies premise rules to specific entities, relations, or textual cues in the meme (e.g., "Check entity X for Contains-DiscriminatoryLanguage using the premise rule Y"). The sequence ends with a logical synthesis step, which aggregates all intermediate results into a final decision according to formal definitions. For example, the final classification can be expressed as Offensive $\Leftrightarrow$ ContainsHateSpeech $\vee$ ContainsDiscriminatoryLanguage $\vee$ ContainsVulgarity $\vee$ MocksOrBelittles, which means that if any of the previous checks is true, the meme is labeled harmful. This structured, rule-grounded plan ensures that reasoning is interpretable, traceable, and directly executable by the Solver.

### 3.4.3 SOLVER

The Solver executes the step-by-step reasoning plan $I$ under multimodal grounding. Specifically, a VLM is employed to verify each step against both textual and visual evidence, ensuring that the reasoning remains faithful to the original meme content.

Given the merged representations $(P_c, Q_c)$, the plan $I$, and the target image $Q_i$, the Solver processes each step $i_j \in I$ sequentially. For each step, it produces a record:

$$D = \langle d_1, d_2, \ldots, d_\ell \rangle \tag{7}$$

where each $d_j$ corresponds to the step $i_j$ in the plan $I$ and specifies the applied premise rule, the relevant entities or contexts, and the intermediate decision with supporting evidence, thereby enabling transparency and explainability.

Finally, the Solver aggregates the trace $D$ to issue the overall judgment: $J \in \{0, 1\}$ where $J = 1$ indicates that the target meme $Q_i$ is harmful. For datasets with multi-class or multi-label annotations, the output can be extended or mapped accordingly, which is discussed in the supplementary material.

## 4 EXPERIMENT

### 4.1 EXPERIMENTAL SETUP

#### 4.1.1 DATASETS

We evaluated MEMORACLE on four widely used meme understanding benchmarks. We first employ the binary-label Facebook Hateful Memes (FHM) dataset (Kiela et al., 2020) and the MultiOff

| Method | Harmful or Not | | | Harmfulness Level | Target Type |
|---|---|---|---|---|---|
| | ACC | BACC | MCC | M-F1 | M-F1 |
| **Open-source LMM (Zero-shot)** | | | | | |
| LLaVA-1.5-13B | 42.94 | 54.04 | 11.08 | 25.83 | 2.19 |
| Qwen2.5VL-7B | 67.23 | 53.97 | 18.23 | 35.92 | 3.82 |
| LLaVA-v1.6-Vicuna-13B | 54.80 | 62.06 | 25.26 | 35.03 | 2.95 |
| LLaVA-OneVision-Qwen2-7B | 68.93 | 58.99 | 24.63 | 36.54 | 4.52 |
| InternVL3.5-8B | 64.41 | 69.82 | 38.99 | 42.86 | 12.25 |
| DeepSeek-VL-7B | 68.93 | 58.99 | 24.63 | 36.54 | 4.52 |
| **Reasoning Based Methods** | | | | | |
| MinD | 41.53 | 54.44 | 15.46 | 25.31 | 12.25 |
| Evolver | 65.82 | 52.51 | 11.20 | 37.86 | 4.60 |
| MemOracle | **70.34** | **71.78** | **41.59** | **51.90** | **15.94** |

(a) Performance on HarM-c.

| Method | Harmful or Not | | | Harmfulness Level | Target type |
|---|---|---|---|---|---|
| | ACC | BACC | MCC | M-F1 | M-F1 |
| **Open-source LMM (Zero-shot)** | | | | | |
| LLaVA-1.5-13B | 51.55 | 52.85 | 8.05 | 24.22 | 4.88 |
| Qwen2.5VL-7B | 54.65 | 53.13 | 11.25 | 35.73 | 4.75 |
| LLaVA-v1.6-Vicuna-13B | 55.77 | 56.49 | 14.06 | 39.77 | 10.50 |
| LLaVA-OneVision-Qwen2-7B | 56.62 | 55.16 | 17.29 | 35.57 | 5.26 |
| InternVL3.5-8B | 55.49 | 56.61 | 16.60 | 35.09 | 30.36 |
| DeepSeek-VL-7B | 53.52 | 54.27 | 9.36 | 33.67 | 8.24 |
| **Reasoning Based Methods** | | | | | |
| MinD | 50.42 | 51.99 | 7.63 | 27.77 | 13.92 |
| Evolver | 55.21 | 53.69 | 13.27 | 32.59 | 11.84 |
| MemOracle | **60.28** | **60.65** | **21.70** | **42.88** | **35.96** |

(b) Performance on HarM-p.

Table 2: Performance comparison on HarM datasets: (a) HarM-c with binary harmful classification, harmfulness level, and target type tasks; (b) HarM-p with the same tasks.

dataset for offensive meme detection (Suryawanshi et al., 2020). Both datasets adopt a binary label scheme to indicate whether a meme is harmful. We additionally include the multi-label Harmful Memes dataset HarM (Pramanick et al., 2021a), which consists of two subsets focusing on COVID-19 (HarM-c) and U.S. politics (HarM-p). Beyond binary classification, HarM-c and HarM-p introduce two finer-grained tasks: Harmfulness Level {Not Harmful, Somewhat Harmful, Very Harmful} and Target Type {Individual, Organization, Community, Society}. Detailed dataset statistics and prompt designs are provided in the Appendix.

### 4.1.2 MODEL SETUP

**Zero-shot Vision–Language Models (VLMs).** We benchmark several open-source VLMs in a zero-shot setting, including `LLaVA-1.5-13B` (Liu et al., 2023), `Qwen2.5-VL-7B` (Wang et al., 2024), `LLaVA-v1.6-Vicuna-13B` (Liu et al., 2024), `LLaVA-OneVision-Qwen2-7B`(Li et al., 2024), `InternVL3.5-8B` (Wang et al., 2025) and `DeepSeek-VL-7B` (Lu et al., 2024a). Preliminary evaluations show that `Qwen2.5-VL-7B` delivers relatively balanced performance across datasets, making it a reasonable choice as the default backbone in our experiments.

**Backbone Models.** For the *Parser* and *Solver* modules, which rely on vision–language understanding, we adopt `Qwen2.5-VL-7B` as the backbone VLM. For the *Translator* and *Planner* modules, which focus on symbolic abstraction and reasoning, we employ `Qwen2.5-14B-Instruct` as the LLM. For associative retrieval, we use `CLIP-ViT-B/32` (Radford et al., 2021) to encode images, extracted elements, and textual descriptions into a unified embedding space, and index the vectors with `Milvus` (Guo et al., 2022) to enable efficient large-scale similarity search.

**Reasoning Methods.** We selected two recently proposed reasoning-based detection methods, `MinD` (Liu et al., 2025) and `Evolver` (Huang et al., 2025), as our baselines. To ensure a fair comparison, all methods are instantiated with the same backbone VLM (`Qwen2.5-VL-7B`).

### 4.1.3 EVALUATION METRICS

We report **Accuracy (ACC)**, **Balanced Accuracy (B-ACC)**, and **Matthews Correlation Coefficient (MCC)** as our primary evaluation metrics, jointly reflecting both overall accuracy and robustness to class imbalance. For multi-label tasks such as HarM, we further include **Macro-F1**, which averages performance across classes and thus provides a fairer assessment under skewed label distributions.

## 4.2 MAIN RESULTS

To evaluate MEMORACLE's effectiveness in harmful meme detection, we conduct experiments on the binary-label datasets FHM and MultiOff. As shown in Table 1, MEMORACLE outperforms prior reasoning-based methods (`MinD` and `Evolver`) across all metrics, achieving 66.10 ACC on FHM and 69.80 ACC on MultiOff. In FHM, it achieves the highest ACC, BACC, and MCC, while in MultiOff it attains the best ACC and MCC and ranks second in BACC, slightly behind LLaVA-v1.6-Vicuna-13B.

We further evaluate MEMORACLE on the multi-label HarM-c and HarM-p benchmarks, which include tasks for binary harmful classification, harmfulness level prediction, and target type prediction. As shown in Table 2, MEMORACLE achieves the best performance across all tasks, reaching 60.28 ACC, 60.65 BACC, and 21.70 MCC for binary harmful classification on the HarM-p benchmark, outperforming prior reasoning-based methods (`MinD` and `Evolver`) by substantial margins. For harmfulness level prediction on the HarM-p benchmark, it attains 42.88 Macro-F1, an improvement of 10.29 points over `Evolver`, and for target type prediction, it achieves 35.96 Macro-F1, far surpassing previous reasoning-based approaches and open-source LMMs. These results demonstrate that MEMORACLE consistently excels in both binary and fine-grained multi-label tasks, highlighting its superior reasoning capabilities.

Figure 4 further illustrates how associative retrieval and symbolic reasoning enable our framework to capture implied harmful intent that backbone VLMs often miss. Taken together, these results confirm that combining symbolic reasoning with associative retrieval allows MEMORACLE to consistently surpass VLM baselines and state-of-the-art reasoning-based methods across diverse tasks.

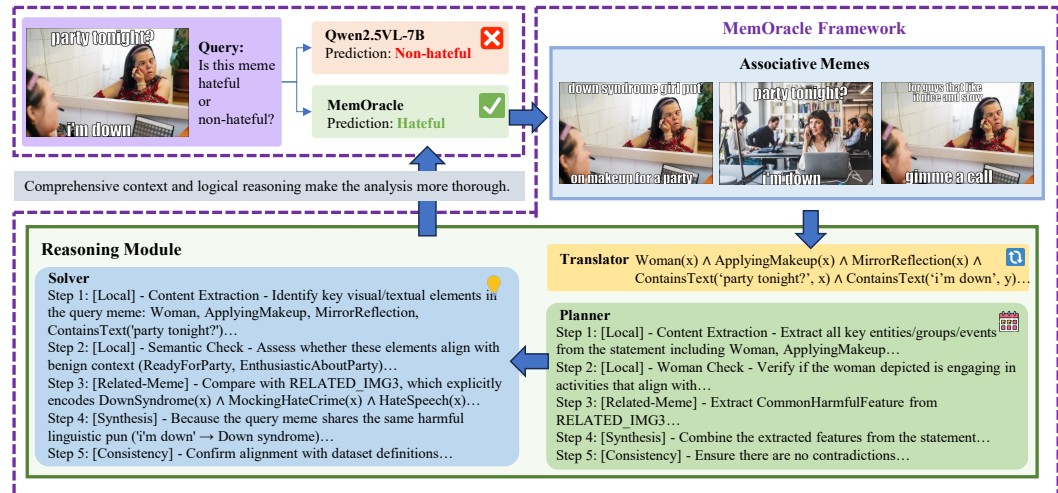

Figure 4: Qualitative examples of MEMORACLE and the vanilla Qwen2.5-VL-7B. Additional cases are provided in the Appendix.

### 4.3 ABLATION STUDY

#### 4.3.1 ASSOCIATIVE RETRIEVAL

This subsection investigates the impact of the number of retrieved memes $k$ on MEMORACLE's accuracy, aiming to understand how retrieval size affects reasoning performance. As shown in Figure 5a, accuracy exhibits a non-monotonic trend over $k \in [0, 10]$: starting at 68.93 with no retrieval ($k = 0$), dropping to 67.23 at $k = 1$, recovering to 68.08 at $k = 2$, and peaking at 70.90 for $k = 3$. Performance remains near the peak for $k = 4$–5 (70.34) before gradually declining. Two opposing effects explain this trend. For small $k \leq 2$, the retrieved evidence is sparse and often idiosyncratic, failing to reveal stable harmfulness cues and sometimes distracting the reasoning module, which accounts for the initial drop. As $k$ increases to a moderate range ($k \approx 3 \sim 6$), associative evidence becomes sufficiently diverse to support abstraction, producing the best results. Beyond that, overly long contexts and the accumulation of weakly related items introduce noise and increase the reasoning load, leading to performance decay. Notably, even without retrieval ($k$=0), MEMORACLE outperforms the `Qwen2.5-VL-7B` baseline by 1.67 percentage points, indicating the effectiveness of our reasoning stages.

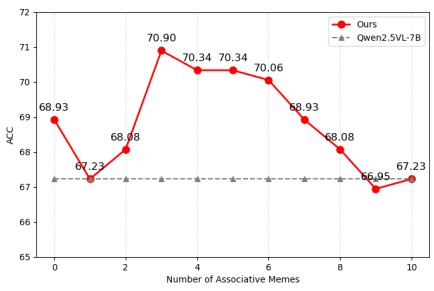

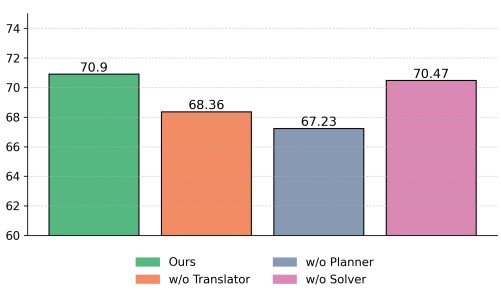

(a) Performance variation with the number of retrieved memes k on HarM

(b) Ablation study.

Figure 5: Impact of removing individual reasoning stages on model accuracy.

#### 4.3.2 EFFECTIVENESS OF SYMBOLIC REASONING STAGES

To assess the contribution of each stage of our reasoning pipeline, we performed ablation studies by removing individual components. As shown in Figure 5b, the complete model achieves an accuracy of 70.90. Removing the Translator reduces the accuracy to 68.36, while removing the Planner results in an even larger drop to 67.23, indicating that these two modules are the most influential. In contrast, removing the Solver only slightly decreases performance to 70.47. Overall, the Planner contributes the largest gain (3.67 points), followed by the Translator (2.54 points). These results validate the effectiveness of our step-by-step planning design for decomposing raw questions into manageable subproblems, while also confirming that symbolic representations and reasoning rules substantially enhance the model's inference capability.

## 5 CONCLUSION

This work investigates the challenging task of hateful meme detection, where harmful intent is often conveyed through subtle multimodal cues, cultural references, and sarcasm. To overcome the lack of structured logic and contextual grounding in prior approaches, we propose MemOracle, a framework that integrates symbolic reasoning with associative retrieval, leveraging historical meme knowledge for context-aware and interpretable detection. Experiments on standard hateful meme benchmarks show that our method achieves significant improvements in both accuracy and robustness over existing baselines. Beyond quantitative gains, MemOracle offers a transparent, logically grounded assessment of multimodal content with explanatory rationales. Looking ahead, we aim to extend its symbolic reasoning and retrieval design to broader multimodal understanding tasks, advancing safer, more responsible, and more explainable AI for online content moderation.

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

## A    THE USE OF LLM

We used a large language model (ChatGPT, OpenAI) solely for English copyediting, including grammar correction, wording and minor stylistic re-writes, and occasional LaTeX formatting help. The model was not used for idea generation, literature search, data collection/annotation, coding, analysis, or producing results. All scientific claims and contributions were written and verified by the authors, and no non-public data were shared with the model. The authors assume full responsibility for the content of the paper.

## B    DATA DETAILS

### B.1    DATASET OVERVIEW

We study four datasets covering three evaluation tasks. **FHM** and **MultiOff** are annotated only for **Task 1**, whereas **HarM-c** and **HarM-p** additionally support **Task 2** and **Task 3**. Dataset sizes (train/test) are listed in Table 3.

| Dataset | Train Samples | Test Samples |
|---------|--------------|--------------|
| FHM | 8,500 | 1,000 |
| MultiOff | 445 | 149 |
| HarM-c | 3,013 | 354 |
| HarM-p | 2,938 | 355 |

Table 3: Dataset statistics (train/test) used in our experiments.

Per-task label definitions are as follows:

**Task 1 (harmfulness, binary):** Decide whether a meme conveys harmful content considering image–text interaction (e.g., slurs, dehumanizing metaphors, threats or disparagement). Labels: 0=*not harmful*, 1=*harmful*.

**Task 2 (harmfulness level, ordinal):** Grade the severity of harmfulness when present; 1 reflects mild/implicit disparagement or sarcasm, while 2 indicates explicit or strongly implied hostility or incitement. Labels: 0=*not harmful*, 1=*somewhat harmful*, 2=*very harmful*.

**Task 3 (influence target type):** Identify the primary target of the harmful implication of the meme conditioned on harmfulness. If Task 1 = 0 or Task 2 = 0, set label of Task 3 to 0 (*none*); Labels: 0=*none*, 1=*individual*, 2=*organization*, 3=*community*, 4=*society*.

| Dataset | Task 1: harmful or Not | | Task 2: Harmful level | | | Task 3: influence target type | | | | |
|---------|------|------|------|------|------|------|------|------|------|------|
|  | 0 | 1 | 0 | 1 | 2 | 0 | 1 | 2 | 3 | 4 |
| FHM | 510 | 490 | – | - | - | - | - | - | - | - |
| MultiOff | 91 | 58 | – | - | - | - | - | - | - | - |
| HarM-c | 230 | 124 | 230 | 21 | 103 | 230 | 119 | 0 | 3 | 2 |
| HarM-p | 184 | 171 | 184 | 159 | 12 | 184 | 81 | 69 | 15 | 6 |

Table 4: Label distribution for each dataset across the three tasks.

### B.2    DATASET-SPECIFIC DESCRIPTIONS AND PROMPT DESIGN

**FHM**    provides a large-scale set of memes. It has 8,500 training examples and 1,000 test examples. and its goal is a binary classification of whether a meme is hateful.

> **Definition**
> A meme is considered hateful if it contains a direct or indirect attack on people based on characteristics, including ethnicity, race, nationality, immigration status, religion, caste, sex, gender identity, sexual orientation, and disability or disease. We define attack as violent or dehumanizing (comparing people to non-human things, e.g. animals) speech, statements of inferiority, and calls for exclusion or segregation. The mocking of hate crimes is also considered hate speech.
>
> **Instruction**
> Please analyze the provided image and respond ONLY with a JSON object in the following format:
> {"is_hateful": <boolean>}
> "is_hateful": Set to true if the image is harmful according to the definition, otherwise false.

Figure 6: Definition and instructions for FHM.

**MultiOff** contains 445 train and 149 test memes collected from Kaggle (2016 US election) and social media platforms, reflecting real-world political memes and emphasizing both image and text as context.

> **Definition**
> A meme is considered offensive if it contains content that is likely to offend, insult, or upset people based on factors such as race, ethnicity, religion, gender, sexual orientation, political views, or other personal characteristics. Offensive content may include offensive speech, discriminatory language, vulgarity, or content that mocks or belittles specific groups or individuals.
>
> **Instruction**
> Please analyze the provided image and respond ONLY with a JSON object in the following format:
> {"is_offensive": <boolean>}
> "is_offensive": Must be `true` if the image is offensive according to the definition, otherwise `false`.

Figure 7: Definition and instructions for MultiOff.

**HarM** includes two datasets: Harm-C (3,544 memes) and Harm-P (3,552 memes) are real-world meme datasets on COVID-19 and US politics, collected from web and social media, deduplicated and filtered, and annotated by experts for harm intensity and target type (individual, organization, community, society).

> **Definition**
> A multimodal unit consisting of an image and embedded text that has the potential to cause harm to an individual, an organization, a community, or society.
>
> **Instruction**
> Please analyze the provided image and respond ONLY with a JSON object in the following format:
> {"harm_level": <string>, "target_type": <string>}
> "harm_level": Must be one of ["not harmful", "somewhat harmful", "very harmful"].
> "target_type": Must be one of ["individual", "organization", "community", "society"].
> If "harm_level" is "not harmful", then "target_type" should be "none".

Figure 8: Definitions and instructions shared by HarM-c and HarM-p.

## C  HYPERPARAMETER IN RETRIEVAL STAGE

In the retrieval stage, we combine three modalities: image embeddings, elements, and generated textual descriptions. Let $\mathbf{v}$, $\mathbf{s}$, and $\mathbf{t}$ denote their respective representations. We form a fused vector

$$\mathbf{r} = \lambda_0 \mathbf{v} + \lambda_1 \mathbf{s} + \lambda_2 \mathbf{t}, \quad \text{with } \lambda_0 + \lambda_1 + \lambda_2 = 1,$$

and use $\mathbf{r}$ to compute retrieval similarity.

Table 5 lists five representative settings. **S1** uses vision only; **S2** and **S3** combine vision with symbolic or textual cues; **S4** assigns approximately equal weights to all modalities; **S5** relies purely on non-visual semantics (element + description).

Empirically, **S5** attains the best accuracy (71.75), while **S4** is a strong second (70.34). We hypothesize that harmfulness is primarily conveyed by semantic information including who is targeted, how,

and in what context, which are better captured by symbolic parses and generated descriptions than by appearance alone. Manual inspection of retrieved memes supports this view: memes that are visually similar can encode opposite meanings due to overlaid text or subtle symbolic cues, which can mislead vision-dominant retrieval (e.g., **S1**–**S3**). Overall, down-weighting the image channel (small $\lambda_0$) while allocating most weight to $\lambda_1$ and $\lambda_2$ tends to yield stronger retrieval for downstream reasoning.

Table 5: Parameter settings for associative retrieval with three modalities. $\lambda_0$, $\lambda_1$, and $\lambda_2$ denote the weights of image, symbolic elements, and generated descriptions, respectively.

| Setting ID | $\lambda_0$ (Image) | $\lambda_1$ (Elements) | $\lambda_2$ (Description) | ACC |
|:---:|:---:|:---:|:---:|:---:|
| S1 | 1.0 | 0.0 | 0.0 | 69.77 |
| S2 | 0.5 | 0.5 | 0.0 | 68.93 |
| S3 | 0.5 | 0.0 | 0.5 | 67.80 |
| S4 | 0.33 | 0.33 | 0.34 | 70.34 |
| S5 | 0 | 0.5 | 0.5 | **71.75** |

# D  SCALING LANGUAGE MODELS

We conducted a controlled study to examine how scaling LLM affects performance within our framework. Here, the VLM is kept fixed as the perceptual backbone, while the LLM responsible for reasoning is varied in size. To further test the generality of our approach, in addition to Qwen2.5-VL-7B used in the main experiments, we also include InternVL3.5-8B, another VLM that performs well in baseline evaluations. Table 6 reports the results on the HarM dataset across all three tasks.

We observe that introducing even a 7B LLM substantially improves performance over the VLM-only baseline, highlighting the effectiveness of symbolic reasoning. While model size has limited impact on the relatively easier binary harmful classification task, larger LLMs consistently yield gains on more fine-grained tasks. In particular, the 14B and 32B models provide notable improvements on harmfulness level prediction and target type identification, indicating that stronger reasoning capacity better supports nuanced inference.

| Method | | Harmful or Not | | | Harmfulness Level | Target type |
|:---|:---|:---:|:---:|:---:|:---:|:---:|
| VLM | LLM | ACC | BACC | MCC | M-F1 | M-F1 |
| Qwen2.5VL-7B | – | 67.23 | 53.97 | 18.23 | 35.92 | 3.82 |
| Qwen2.5VL-7B | Qwen2.5-7B | 67.80 | 67.78 | 34.13 | 47.68 | 11.02 |
| Qwen2.5VL-7B | Qwen2.5-14B | **70.34** | **71.78** | **41.59** | **51.90** | 15.94 |
| Qwen2.5VL-7B | Qwen2.5-32B | 68.08 | 70.79 | 39.73 | 46.98 | **16.01** |
| InternVL3.5-8B | – | 64.41 | 69.82 | 38.99 | 42.86 | 12.25 |
| InternVL3.5-8B | Qwen2.5-7B | **68.93** | 69.21 | 36.83 | 47.63 | 12.67 |
| InternVL3.5-8B | Qwen2.5-14B | 67.80 | **72.62** | **43.93** | **50.82** | 16.70 |
| InternVL3.5-8B | Qwen2.5-32B | 68.36 | 71.19 | 40.52 | 48.63 | **17.27** |

Table 6: Performance of API-based models on the HarM dataset across three tasks.

# E  SYMBOLIC REASONING MODULE DESIGN

Section 3.4 describes our symbolic reasoning pipeline. Due to space limitations, the main text illustrates only the Translator. Here we present the visualizations of the remaining two components—the *Planner* and *Solver*—referenced in Section 3.4.

**Prompt**                                                                  Large-Language Model

"Please derive a step-by-step plan using the First-OrderLogic rule for determining the conclusion based on the context."

**Input**

"Symbolic Definition": ["Attack(x): x is an attack based on characteristics", "Violent(x): x is violent speech or actions", "Dehumanizing(x): x compares people to non-human things"]

"Translated Premise": { "dataset_content": [ "∀x (Attack(x) ↔ (Violent(x) ∨ Dehumanizing(x) ∨ Inferiority(x) ∨ Exclusion(x) ∨ Segregation(x) ∨ MockingHateCrime(x)))…"],

"related_img1": ["∀x (TwoPanelLayout(x) ∧ Man(x) ∧ WearsSuit(x) ∧ Bald(x) ∧ TextTop('its their character', x)…"],

"related_img2": ["∀x (SinglePanelLayout(x) ∧ Man(x) ∧ WearsSuit(x) ∧ HoldsMicrophone(x) ∧ Smiling(x)…"]}

"Translated Statement": {"query_meme": {"Man(x) ∧ WearsSuit(x) ∧ Man(y) ∧ Bald(y)…"},

"question": "ContainsHateSpeech(x)"}

**Output**

"Step by Step Plan": ["Step 1: [Local] - Content Extraction - Parse symbols from input: two male portraits; TextTop('its their character')…",

"Step 2: [Local] - Symbol Mapping - From Symbolic Definition and Premise: Attack(x) ↔ (Violent(x) ∨ Dehumanizing(x) ∨ Inferiority(x) ∨ Exclusion(x) ∨ Segregation(x)…",

"Step 3: [Local] - Protected Attribute - Detect MentionsProtectedAttr(x, color) from 'color'. Using: Modus Ponens.",

"Step 4: [Related-Meme] - Retrieve related_img* by cues {'character', 'color', 'skin', 'race'}; compute CommonHarmfulFeature as intersection/majority over…",

"Step 5: [Synthesis] - If CommonHarmfulFeature = ∅ and no harmful predicates hold locally, infer ¬Attack(x); else Attack(x)…",

"Step 6: [Decision] - Define ContainsHateSpeech(x) ↔ (Attack(x) ∧ MentionsProtectedAttr(x, _)); output Harm Level and Target Type. Using: Modus Ponens + Consistency Check."]

Figure 9: Illustration of the Planner Stage

**Prompt**                                                                  Vision-Language Model

"Your task is to carry out a logical solution based on the Step-By-Step Plan and input image."

**Input**

"img": "img/42953.png"

"Symbolic Definition": ["Attack(x): x is an attack based on characteristics", "Violent(x): x is violent speech or actions", "Dehumanizing(x): x compares people to non-human things"]

"Translated Premise": { "dataset_content": [ "∀x (Attack(x) ↔ (Violent(x) ∨ Dehumanizing(x) ∨ Inferiority(x) ∨ Exclusion(x) ∨ Segregation(x) ∨ MockingHateCrime(x)))…"],

"related_img1": ["∀x (TwoPanelLayout(x) ∧ Man(x) ∧ WearsSuit(x) ∧ Bald(x) ∧ TextTop('its their character', x)…"],

"related_img2": ["∀x (SinglePanelLayout(x) ∧ Man(x) ∧ WearsSuit(x) ∧ HoldsMicrophone(x) ∧ Smiling(x)…"]}

"Translated Statement": {"query_meme": {"Man(x) ∧ WearsSuit(x) ∧ Man(y) ∧ Bald(y)…"},

"question": "ContainsHateSpeech(x)"}

"Step by Step Plan": ["Step 1: Identify and analyze the symbolic definitions provided in the Symbolic Definition section to understand the domain concepts…",

"Step 2: …"]

**Output**

"img": "img/42593.png"

"label": 0,

"solution": ["Step 1: [Local] - Content Extraction - TwoPanel; Man(x), Man(y); TextTop('its their character')…",

"Step 2: [Local] - Protected Attribute - MentionsProtectedAttr(m, color) detected (skin color)…",

"Step 3: [Local] - Harm Predicates - EqualityMessage(m) ⇒ ¬Violent(m) ∧ ¬Dehumanizing(m) ∧ ¬Inferiority(m) ∧ ¬Exclusion(m)…",

"Step 4: [Related-Meme] - Majority over related_img* shows equality/anti-discrimination…",

"Step 5: [Synthesis] - Attack(m) ↔ (Violent ∨ Dehumanizing ∨ Inferiority ∨ Exclusion ∨ Segregation ∨ MockingHateCrime)…",

"Step 6: [Decision] - ContainsHateSpeech(m) = Attack(m) ∧ MentionsProtectedAttr(m, color) ⇒ false…" ]

Figure 10: Process Visualization of covid_memes_5560 form HarM

# F PROCESS VISUALIZATION

To provide a more intuitive understanding of how our framework operates, we visualize several representative examples that illustrate the intermediate steps in both the retrieval and reasoning stages. The visualizations show how relevant memories are retrieved across different modalities and how the retrieved information is subsequently integrated into the reasoning process. These examples are intended to complement the quantitative results by offering a step-by-step view of the model's internal workflow.

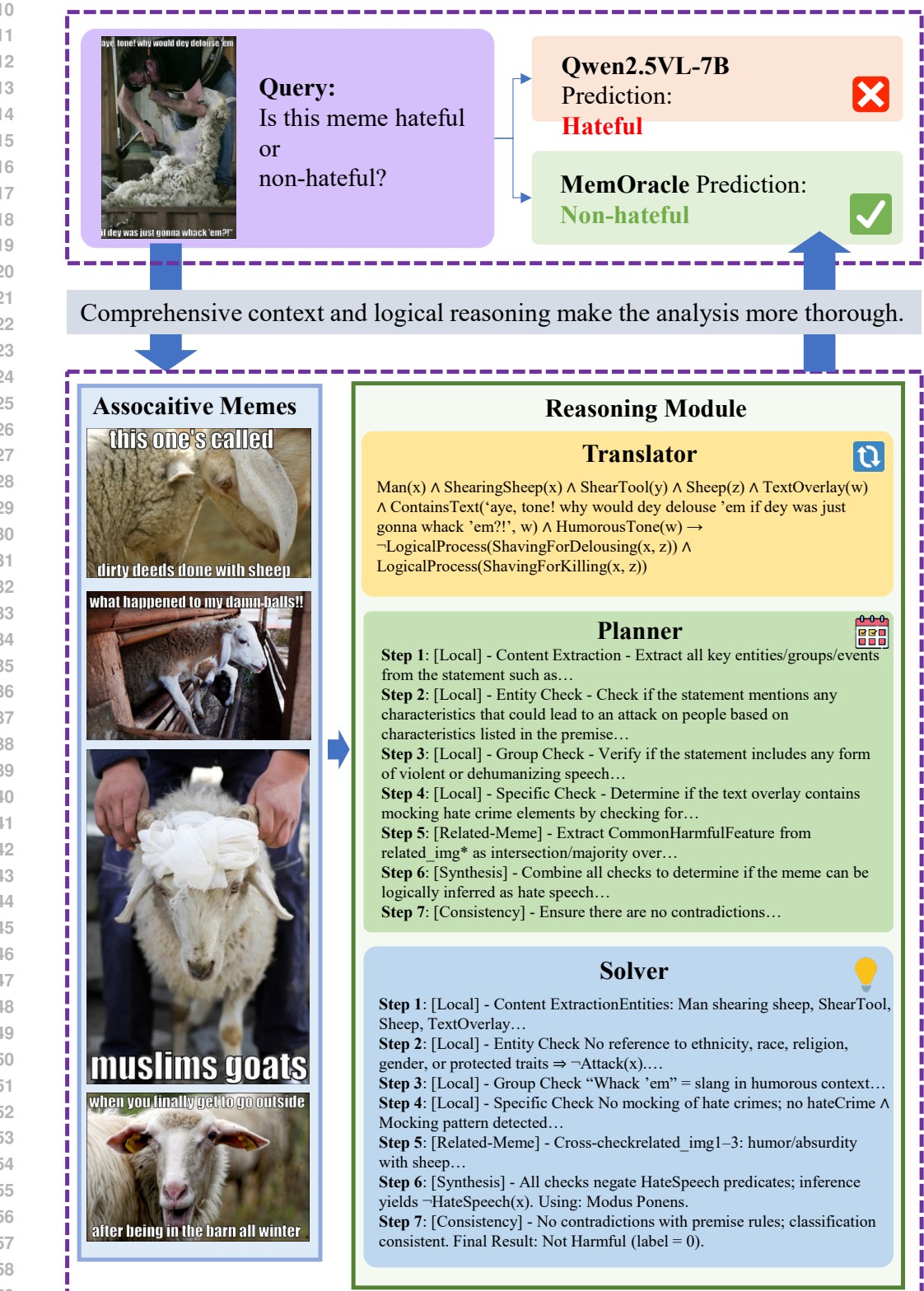

Figure 11: Process Visualization of 03164 form FHM

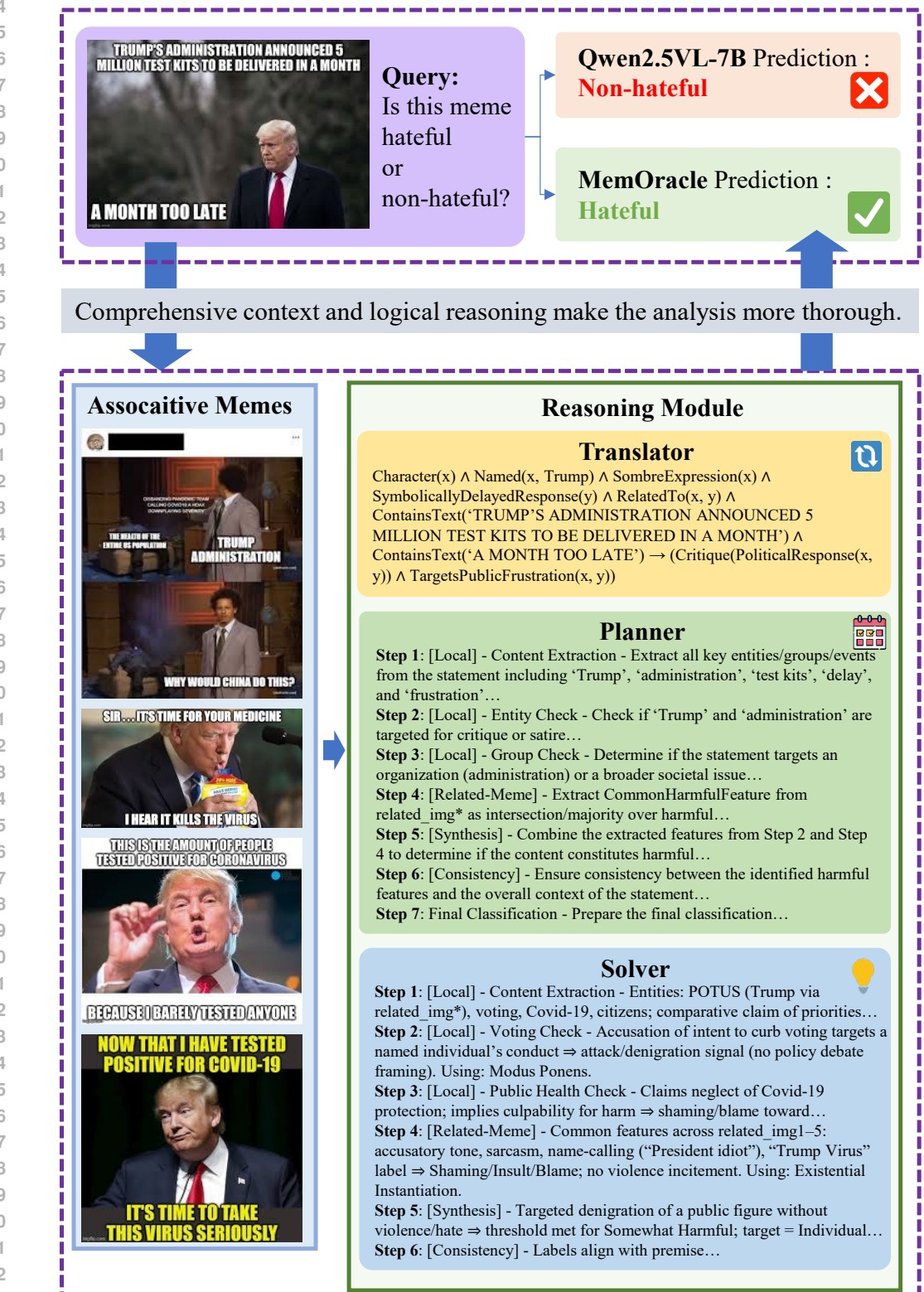

Figure 12: Process Visualization of covid_memes_5560 form HarM

