# OpenReview forum: "MemOracle: Symbolic Reasoning with Associative Retrieval for Harmful Content Detection"
_ICLR.cc/2026/Conference — Submitted to ICLR 2026_

### Official Review · Reviewer_tgpK · 2025-10-18

**Soundness:** 2
**Presentation:** 2
**Contribution:** 2
**Rating:** 2
**Confidence:** 4

**Summary:**

This paper present MEMORACLE to tackle meme classification through retrieval and reasoning. MEMORACLE is a framework combining the extraction of textual features through VLM, performing similarity-based dense vector retrieval through CLIP, and then performing symbol-based Chain-of-Thought prompting through translator, planner, and solver.  The CoT reasoning is based on the retrieved similar examples, acting as demonstrating associative examples. The purposed method is evaluated on various hateful meme datasets, demonstrating the effectiveness of the method.

**Strengths:**

- The symbolic CoT reasoning pipeline is novel for hateful meme detection and reasoning.
- The system produce interpretable reasoning and detection results.

**Weaknesses:**

- The authors claim that their method achieves state-of-the-art performance. However, this assertion is not supported by the reported results, which lag behind several existing approaches. For instance, PromptHate[1] reports accuracies of 72.98% on FHM and 84.47% on HarM-c, compared to MEMORACLE’s 66.10% and 70.34%. Similarly, RGCL[2] achieves 78.8% and 87.0% on the same datasets, respectively. This is almost 20% of accuracy difference. I would argue such binary classification system with sub 70% accuracy would be unusable. These methods also rely on much smaller backbones such as CLIP, rather than expensive reasoning-based inference systems with much larger LLM. Although those models are fine-tuned, the authors’ claim of state-of-the-art performance remains overstated. If the authors intended to refer to the zero/few-shot setting, that should be clearly specified. Even then, several recent methods perform comparably or better under such conditions. For example, LOREHM[3] achieves 70.2% and 74.6% on FHM and HarM-c, respectively, using a similar retrieval-plus-reasoning pipeline. More recent works, such as RA-HMD and U-CoT+, further improve upon these results (though these may be considered concurrent).

- Furthermore, BridgeMod[4] demonstrates that even a simple BM25-based retrieval (for few-shot demonstrating examples retrieval, similar as the associative retrieval proposed) combined with naive LLM classification, without explicit symbolic reasoning, can achieve comparable results (66.0% with Mistral-7B and 65.4% with Qwen2-7B on FHM). Together with the stronger results of LOREHM, which also integrates CLIP-based retrieval and text-based chain-of-thought reasoning, this raises questions about the added value of MEMORACLE’s symbolic reasoning pipeline.

- The authors state that the framework “produces judgments that are both logically consistent and faithful to the original multimodal content.” I would recommend evaluating this claim using an external LLM-as-a-judge approach to objectively assess logical consistency and faithfulness. Furthermore, since the paper emphasizes enhancing the interpretability of hateful meme detection, it would be valuable to include quantitative evaluations that support this claim (e.g., measuring explanation quality, agreement with human rationale, or other interpretability metrics).

- The paper includes only one and a half pages of citations, which seems insufficient for the scope of the work. The authors should consider adding several relevant and recent references, particularly those referred above. Incorporating these works would improve the paper’s positioning and strengthen its connection to prior research. I would also suggest improving the related work sections.




[1]Rui Cao, Roy Ka-Wei Lee, Wen-Haw Chong, and Jing Jiang. 2022. Prompting for Multimodal Hateful Meme Classification. In Proceedings of the 2022 Conference on Empirical Methods in Natural Language Processing, pages 321–332, Abu Dhabi, United Arab Emirates. Association for Computational Linguistics.

[2] Jingbiao Mei, Jinghong Chen, Weizhe Lin, Bill Byrne, and Marcus Tomalin. 2024. Improving Hateful Meme Detection through Retrieval-Guided Contrastive Learning. In Proceedings of the 62nd Annual Meeting of the Association for Computational Linguistics (Volume 1: Long Papers), pages 5333–5347, Bangkok, Thailand. Association for Computational Linguistics.

[3] Jianzhao Huang, Hongzhan Lin, Liu Ziyan, Ziyang Luo, Guang Chen, and Jing Ma. 2024. Towards Low-Resource Harmful Meme Detection with LMM Agents. In Proceedings of the 2024 Conference on Empirical Methods in Natural Language Processing, pages 2269–2293, Miami, Florida, USA. Association for Computational Linguistics.

[4] Ming Shan Hee, Aditi Kumaresan, and Roy Ka-Wei Lee. 2024. Bridging Modalities: Enhancing Cross-Modality Hate Speech Detection with Few-Shot In-Context Learning. In Proceedings of the 2024 Conference on Empirical Methods in Natural Language Processing, pages 7785–7799, Miami, Florida, USA. Association for Computational Linguistics.

**Questions:**

- Regarding inference cost, the system seems to combine multiple stages using LMM and LLM, have you measured the inference cost? It would be helpful to understand the system’s computational efficiency relative to existing baselines.


- In Appendix D, the authors compare performance across different language models. I am curious whether the overall performance bottleneck could stem from the limited capacity of the LMMs used in the visual encoding stage. Since all visual information is condensed into textual descriptions, the richness of these representations might depend heavily on the vision model’s capability and the image-to-text prompting strategy. Have the authors explored using stronger visual encoders or refining the image-to-text prompts to improve this step? This seems plausible given that closed-source systems like GPT-4o achieve higher zero-shot accuracy on datasets such as FHM (over 70%) without explicit reasoning, suggesting that the visual understanding and grounding ability of the more capable model could play a key role.

---

### Official Review · Reviewer_rFu3 · 2025-10-29

**Soundness:** 2
**Presentation:** 2
**Contribution:** 2
**Rating:** 4
**Confidence:** 4

**Summary:**

This paper proposes MemOracle, a framework for hateful meme detection that integrates multimodal retrieval with a symbolic reasoning pipeline (Translator, Planner, Solver). The core idea of enhancing a VLM with structured, logical reasoning is novel and has merit.

**Strengths:**

Addressing a critical and timely challenge in online content moderation by tackling the nuanced problem of hateful meme detection through multimodal reasoning. Introducing a novel framework that effectively integrates structured symbolic reasoning with associative retrieval, enhancing both detection performance and model interpretability.

**Weaknesses:**

1. The manuscript does not adhere to standard academic formatting conventions, which significantly detracts from its professionalism and readability.
- As noted, table captions must be placed above the table, not below. All figures and tables must have descriptive captions.
- The stray hyphen (`-`) before the abstract is unacceptable.
Inconsistent and enlarged fonts in tables and figures suggest a lack of care in preparation.
- The authors must meticulously proofread the entire document and strictly conform to the conference's style and formatting guidelines before any re-submission.

2. The literature review fails to properly contextualize this work. The review largely ignores relevant work from the last 3-4 years. The field of explainable and reasoning-based meme detection is more mature than portrayed. The paper claims a key contribution in "interpretability" through symbolic reasoning. However, this claim is not substantiated against existing work that also provides explanations. The authors must thoroughly survey and differentiate their approach from these existing methods for providing explanations and logical reasoning in multimodal hate speech detection.

3. The benchmarks primarily compare against smaller open-source LMMs (up to 13B parameters). This is insufficient. To establish a strong baseline, the authors should include comparisons with: Models like GPT-4V, GPT-4o, and Gemini 1.5/2.0 Pro, which are known to possess strong multimodal reasoning capabilities. Models such as LLaVA-1.5-34B, or other VLMs in the 30B+ parameter range.

**Questions:**

1. As above
2. Is there any justification for the selection of backbone models, such as Qwen2.5 and CLIP-ViT-B/32 (associative retrieval), in the methods used?

---

### Official Review · Reviewer_KTB5 · 2025-10-30

**Soundness:** 3
**Presentation:** 3
**Contribution:** 2
**Rating:** 2
**Confidence:** 5

**Summary:**

The paper presents a coherent, timely pipeline for multimodal hateful meme detection: it parses the meme, retrieves semantically related “premises,” and reasons over them using a three-stage LLM module. This is a conceptual step beyond treating memes as single-instance classification, and the authors show that both retrieval and the symbolic planner actually matter through ablations. However, unresolved weaknesses remain. First, results are all single-number reports with no statistical significance or variance, so the claimed gains over MinD/Evolver are not yet defensible. Second, the retrieval story is shown only within the same meme pools used for evaluation, so the claimed “associative retrieval” has not yet been demonstrated to generalize to out-of-pool or cross-domain memes. On top of that, the pipeline is heavy, multiple large models plus a vector DB, with no latency/cost section, which weakens the case for deployability in real moderation workflows.

**Strengths:**

S1: Retrieval-augmented, modular architecture that unifies VLM perception with LLM-based symbolic reasoning for multimodal hateful meme detection, enabling more context-aware decisions.

S2: Consistent, strong results across four public meme/harm datasets (FHM, MultiOff, HarM-c, HarM-p) against strong VLM and recent reasoning baselines, demonstrating breadth of applicability within this subset.

S3: Ablations that remove individual reasoning stages and show clear performance drops, supporting the claim that symbolic planning is an essential part of the gains.

S4: Qualitative examples that illustrate how retrieval + symbolic reasoning can correct VLM errors and provide human-interpretable rationales, which is valuable for moderation workflows.

**Weaknesses:**

W1: Reported improvements are single-run numbers; despite having multiple datasets and tasks, the absence of statistical significance or variance analysis makes the gains less conclusive.

W2: The pipeline depends on several large components (VLM, LLM, CLIP, vector DB); although the modularity is attractive (see S1), the paper does not provide an efficiency, cost, or latency profile, so deployability in real-time moderation remains unclear.

W3: Retrieval is built from the same meme datasets used for evaluation; while the current retrieval is shown to matter, this leaves open whether the “associative retrieval” idea holds on out-of-pool or more diverse/multilingual memes.

W4: Symbolic reasoning is defined through LLM prompting, but there is no robustness study to prompt changes or noisy parses, so reliability in realistic, noisy inputs is not established.

W5: The approach explicitly assumes access to historical or related content and even notes possible bias and the need for a buffer period, but there is no reported performance for “no/low-neighbor” settings.

W6: Even though the authors provide ablations over k and over reasoning stages, the paper does not fully disentangle how much of the gain comes from retrieval quality vs. symbolic template/planner quality, limiting attribution.

W7: The method relies on accurate in-meme text capture, but there is no measurement of degradation under poor OCR, stylized fonts, or low resolution.

**Questions:**

Q1: Would the authors consider testing and benchmarking against newer and bigger VL models?

---

### Official Review · Reviewer_ceAb · 2025-11-01

**Soundness:** 2
**Presentation:** 2
**Contribution:** 2
**Rating:** 4
**Confidence:** 4

**Summary:**

The paper proposes MEMORACLE, a retrieval-augmented multimodal framework that combines a vision–language parser, associative retrieval (Milvus + CLIP embeddings), and a three-stage symbolic reasoning pipeline to detect hateful/harmful memes. The system translates multimodal content into first-order logic (FOL) style predicates, retrieves K nearest “associative” memes from a vector memory, and uses an LLM/VLM combination to produce step-by-step rationales and a final binary/multi-label decision. Experiments report improvements over recent reasoning baselines (MinD, Evolver) and several open LMM baselines on FHM, MultiOff and HarM variants. Key implementation details and ablations (retrieval size k, component removal, modality weighting) are provided in the Appendix.

**Strengths:**

1.	Addresses an important and practical problem (hateful/harmful meme detection) where multimodal, contextual reasoning is genuinely needed.
2.	The pipeline is comprehensive and evaluated on several standard datasets (FHM, MultiOff, HarM variants), with multiple metrics (ACC, BACC, MCC, Macro-F1) reported.
3.	Ablation experiments (retrieval size k, removal of Translator/Planner/Solver stages, modality weighting) are included and help identify which modules contribute most to performance.

**Weaknesses:**

1. The Translator claims to output FOL expressions, but the paper lacks a formal grammar, typing rules, or a description of how ambiguous natural language/image inputs map to logical atoms, predicates, and quantifiers. This gap prevents independent verification and may hide brittle heuristics used in practice. Provide a formal specification (or pseudo-code) of the translation algorithm and discuss failure cases.

2. The Planner and Solver are described at a high level (step sequences, Modus Ponens usage), but the system’s treatment of contradictions, uncertain evidence, or conflicting retrieved premises is unspecified. How are contradictory retrieved memes reconciled? Is there weighting or belief tracking? These are crucial for symbolic reasoning claims.

3. Report standard deviations across multiple runs, provide random seeds, and include statistical tests to support claims like “outperforms by up to 9%”. Without confidence intervals or p-values the numerical claims are weak.

4. Ablations show the Planner is important, but do not isolate whether gains come from (a) better parse quality, (b) retrieval content, or (c) LLM prompt phrasing. I recommend: (i) ablate quality of Translator outputs (oracle symbolic inputs vs. automatic), (ii) evaluate retrieval relevance quantitatively (precision@k) and (iii) ablate prompt templates and LLM temperature/decoding.

5. The paper claims interpretable step-by-step rationales. Provide a human evaluation (annotator agreement / faithfulness / helpfulness) to substantiate interpretability claims—automatic metrics are insufficient.

6. Table 1/2 placement, spacing inconsistencies (`first-order logic(FOL)` vs `first-order logic (FOL)`), and figure alignment problems reduce readability. Fix figure subpanel alignment, ensure consistent typography, and place tables adjacent to first references.

**Questions:**

1. Please provide the formal specification (grammar or schema) used by the Translator to produce FOL atoms and the mapping rules for common image/text constructs (e.g., panels, dialogues, sarcasm markers). If this uses learned prompts, provide the exact prompt templates and examples (oracle vs automatic).

2. How does the Planner/Solver handle contradictory retrieved premises? Is there a ranking/weighting or conflict-resolution mechanism? Please describe the concrete merge (⊕) operator and provide pseudocode for the Planner/Solver execution.

3. Provide variance estimates (std. dev) for the main reported metrics and indicate number of random seeds / runs used. Were the retrieval indices or LLMs deterministic across runs?

4. Can you show examples where MEMORACLE produces an incorrect rationale but still the correct final label (and vice versa)? Such failure analyses would clarify whether the rationales are faithful or post-hoc.

---

### Meta-Review · Area_Chair_vJaP · 2026-01-06

**Summary:**

Although this paper deals with an important and timely topic, several concerns were raised by reviewers. These include lack of methodological and formal clarity (underspecified symbolic reasoning claims, for example), weak evaluation rigour and statistical validity where nearly all reviewers flag weaknesses in evaluation (single-run results only, with no variance, confidence intervals, or statistical significance tests, for example), unsubstantiated interpretability claims (for example, no failure analysis contrasting correct labels with incorrect rationales (or vice versa) and concerns that explanations may be post-hoc), as well as significant formatting and style violations, inconsistent typography, and table/figure placement issues.

**Reviewer Concerns:**

The authors did not respond to any of the reviewers concerns

**Reviewer Scores:**

The paper received 2 x rating 2 (reject) and 2x rating 4 (marginally below the acceptance threshold)

---

### Decision · Program_Chairs · 2026-01-26

Reject